# Differential Distribution of RBPMS in Pig, Rat, and Human Retina after Damage

**DOI:** 10.3390/ijms21239330

**Published:** 2020-12-07

**Authors:** Xandra Pereiro, Noelia Ruzafa, J. Haritz Urcola, Sansar C. Sharma, Elena Vecino

**Affiliations:** 1Department of Cell Biology and Histology, Experimental Ophthalmo-Biology Group (GOBE), University of the Basque Country UPV/EHU, 48940 Leioa, Vizcaya, Spain; xandra.pereiro@ehu.eus (X.P.); noelia.ruzafa@ehu.eus (N.R.); javieraritz.urcola@ehu.eus (J.H.U.); sansar_sharma@nymc.edu (S.C.S.); 2Department of Ophthalmology, Araba University Hospital, 01009 Vitoria, Alava, Spain; 3Department of Anatomy and Cell Biology, New York Medical College, Valhalla, NY 10595, USA

**Keywords:** RNA binding protein with multiple splicing, retinal ganglion cells, development, inner plexiform layer, ganglion cell layer

## Abstract

RNA binding protein with multiple splicing (RBPMS) is expressed exclusively in retinal ganglion cells (RGCs) in the retina and can label all RGCs in normal retinas of mice, rats, guinea pigs, rabbits, cats, and monkeys, but its function in these cells is not known. As a result of the limited knowledge regarding RBPMS, we analyzed the expression of RBPMS in the retina of different mammalian species (humans, pigs, and rats), in various stages of development (neonatal and adult) and with different levels of injury (control, hypoxia, and organotypic culture or explants). In control conditions, RBPMS was localized in the RGCs somas in the ganglion cell layer, whereas in hypoxic conditions, it was localized in the RGCs dendrites in the inner plexiform layer. Such differential distributions of RBPMS occurred in all analyzed species, and in adult and neonatal retinas. Furthermore, we demonstrate RBPMS localization in the degenerating RGCs axons in the nerve fiber layer of retinal explants. This is the first evidence regarding the possible transport of RBPMS in response to physiological damage in a mammalian retina. Therefore, RBPMS should be further investigated in relation to its role in axonal and dendritic degeneration.

## 1. Introduction

RNA binding protein with multiple splicing (RBPMS), also called HERMES, is a conserved RNA binding protein with a single RNA recognition motif (RRM). RBPMS is expressed in the retina, heart, liver, and kidney, and to a lesser degree in the cerebellum, cerebral cortex, lung, and small intestine. RBPMS is only abundantly expressed in retinal ganglion cells (RGCs) in the retina and was used as an RGC specific marker to differentiate RGCs from other retinal neurons and non-neuronal cells [1,2]. Similarly, Brn3 and Thy-1 have been used to identify RGCs; however, these markers do not label the entire RGC population [3,4]. Recent studies have revealed that RBPMS can label all RGCs in the normal retinas of mice, rats, guinea pigs, rabbits, cats, and monkeys [5,6].

The exact function of RBPMS is unknown; proteins of the RRM family are involved in regulation of pre-mRNA-processing (splicing, capping, and polyadenylation), RNA stability, transport, localization, and regulation of mRNA translation. Many neuronal RRM proteins are expressed in a region-specific manner [7] and are involved in post-transcriptional regulation, which are important for the nerve cells with respect to their maturation and function [8]. Some of them have been shown to play a functional role in the response to guidance cues in both axons [9,10,11] and dendrites [12]. A recent report showed that exogenously expressed RBPMS proteins are localized in both axons and dendrites in culture of cortical neurons [13]. In *Xenopus* and zebrafish RGCs, RBPMS-positive granules were found in the axons of maturing neurons and their function in RGC axons could be associated with arbor and synapse formation [14]. In the RGC-5 retinal cell line that is derived from photoreceptor cells, RBPMS forms cytoplasmic granules and may be transported in elaborating neurites [15]. However, its function in mammalian retinas remains unknown.

In the present study, three different species were used in order to compare the expression of RBPMS: pigs, rats, and humans. The porcine retina is an excellent model for studying diverse retinal processes and diseases as it is phylogenetically close to that of the human and is easily available compared to the ape retina. The pig eye/retina shares many similarities with that of the human [16,17,18]. The porcine retina is even more similar to the human retina than that of other large mammals such as the dogs, goats, or cows [19]. There are no published studies on the expression of RBPMS in the pig retina. On the other hand, rat retinas are commonly used to investigate optic nerve diseases. There is great conservation between rats and mice, and human genomes. The structures of their optic nerve and retina, however, have a number of differences from those of humans: the eyes of rats and mice do not have maculae or foveae. Rats’ eyes are commonly used to study optic nerve injuries [20]. Thus, in the present study, the expression of RBPMS was studied in these three species.

There is evidence to suggest that RBPMS may be important during development, for example, the highest expression level of RBPMS was detected in the myocardial tissues of the developing heart, coincident with the time of appearance of myocardial differentiation markers (Gerber, et al. 1999). In the developing eyes of *Xenopus*, RBPMS transcripts were detected only in RGCs at stage 34 [21]. In addition, as a result of the highest expression of RBPMS in other tissues during development, previous studies suggest that RBPMS could have a role in axon and dendrite formation [14]. Thus, we studied the expression of RBPMS in RGCs in pigs (neonatal and adult) under hypoxic conditions and in a degenerative pig retina model (explants), and we compared the results with other mammalian species: humans and rats.

## 2. Results

To verify the expression of RBPMS in pig RGCs, we labeled the whole-mount retinas with antibodies against Brn3a and against RBPMS to confirm that RBPMS is a good marker for RGCs. The entire population of RGCs that were marked with Brn3a were also marked with RBPMS; however, some RGCs were marked only with RBPMS (Figure 1), showing that RBPMS is a better marker of RGCs than Brn3a.

Furthermore, the eyes were kept for 24 h in a CO_2_-independent medium at 4 °C as a model of hypoxia and the expression of RBPMS in retinas from adult pigs and rats; these were observed in control and under hypoxic conditions. In control conditions, both pig and rat retinal sections RBPMS was localized in the somas of RGCs (Figure 2A,C). However, under hypoxic conditions, the RBPMS was expressed in the inner plexiform layer (IPL), whereas the RGCs had reduced labeling of RBPMS in the somas (Figure 2B,D) in both pig and rat retinas.

In addition, the adult human retina was analyzed to compare the expression pattern of RBPMS to the previously analyzed animal models. RBPMS was expressed in the inner nuclear layer (INL) in human retinal sections and reduced in the somas of RGCs, a pattern similar to pig and rat retinas after hypoxia (Figure 3). This could be due to the fact that the human eyes were obtained 24 h postmortem and could likely be due to the hypoxic effect as compared to retinas collected immediately after death (controls).

As a result of the importance of RBPMS in development, the expression of RBPMS was analyzed in neonatal and adult pig retinas in order to compare if the localization of RBPMS in RGCs could also be different depending on the age of the animal. Adult and pig retinas were analyzed in both control and hypoxic conditions. Differences between the sections from adult and neonatal retinas were not found (Figure 4).

In order to demonstrate that the different pattern of expression of RBPMS was not due to the antibody recognizing a different protein in control and hypoxic retinas, a Western Blot was utilized. In both control and hypoxic pig retinas, the expression of RBPMS was detected (Figure 5). It should be noted that the bands of 20 µg of protein in the control are similar to the band of 40 µg protein of hypoxic retinas, and consequently it seems that there is increased expression of RBPMS in control retinas. These results corroborate that RBPMS has the same molecular weight in control and hypoxic retinas; however, it had different distribution in soma versus dendrites, and therefore different increased levels (Figure 5).

Finally, adult isolated pig retinas were used in organotypic culture as a model of neurodegeneration. Immunocytochemistry was performed to identify RBPMS. In sections of the organotypic retinal at 24 h of culture, the punctate labeling of RBPMS was present in the nerve fiber layer, as we demonstrated when we combined the RBPMS labeling with neurofilament labeling (Figure 6A–I). In the whole mount retinal organotypic cultures, we observed that the RBPMS labeling was punctate in the optic axons (Figure 6L).

## 3. Discussion

RBPMS is a member of the RRM family of RNA-binding proteins and is found in distinct tissues of various vertebrate species from fish to humans. It has been shown that members of the RRM family are involved in the regulation of gene expression at the post-transcriptional level [8]. In the present study, we show that RBPMS have a different localization after damage.

In neurons, proteins drive the function of neuronal synapses. The synapses are distributed throughout the dendritic arbor, often hundreds of micrometers away from the soma. It is still unclear how somatic and dendritic sources of proteins shape protein distribution and respectively contribute to local protein changes during synaptic plasticity. Growing experimental evidence indicates that mRNAs are abundant in dendrites [22,23], and an extensive body of literature shows that local translation plays an important role in many forms of developmental and synaptic plasticity [24,25,26]. The distribution pattern of RBPMS in control pig and rat retinas was similar; it is located in the soma of RGCs. However, after an event of hypoxic damage, RBPMS is translocated to dendrites. RBPMS in human retinas was also located in dendrites. This could be due to the fact that humans’ eyes were received 24 h after they were removed from the donor, similar to our model of hypoxia. It is well known that the pig eye most closely resembles the human eye, with similar size and comparable histological and physiological features; consequently, the same distribution pattern must exist between adult pig and human retinas. Herein, we confirm that RBPMS behaves similarly in the different mammalian species that were analyzed.

There is rather limited knowledge regarding the function of RBPMS: several studies have reported that RBPMS has different localizations in the RGCs in *Xenopus* and zebrafish. It has been shown that RBPMS is present in axons, and blocking it leads to a significant reduction in retinal axon arbor complexity in the tectum, but an increase in the density of presynaptic puncta [14]. Furthermore, in the RGC-5 cell line, RBPMS interacts with other proteins and forms cytoplasmic granules, suggesting that it is transported in neurites. In that study, they showed that RBPMS migrate from the soma to the dendrites, suggesting that RBPMS may have a role in dendrites formation [15]. However, transport of mRNA to distal neuronal processes, including both dendrites and axons, confers a number of potential advantages on the cell. Localizing populations of transcripts far from the soma allows the cell to synthetize proteins in response to local signaling on rapid timescales. mRNA can be transported “ahead of time” in a translationally repressed state, without committing to the functionality of the expressed protein until it is needed [27]. Thus, RBPMS as an RNA binding protein could be implicated in these processes.

To examine other types of damage and retinal survival mechanisms, organotypic retinal explant models were used as a model of neurodegeneration. RGC can be preserved in situ with their normal adjacent retinal architecture, which maintains the environment closer to in vivo conditions. It is known that in rodents, RGCs die at a predictable rate after optic nerve crush and there is a marked reduction in RBPMS protein expression at day 1 after axotomy [28,29,30]. In the present study, in 1-day explants of pig retina, RBPMS is localized in the NFL showing a punctate pattern in the axons of degenerated RGCs after axotomy, suggesting that RBPMS may also be used as a marker for degenerating RGCs. However, recent studies using RNAseq on RGCs axons in vivo [31] have identified a highly complex axonal transcriptome that includes transcripts of multiple classes, similar to the somatodendritic compartment. There is growing evidence that axonal translation is present in developing axons and in axons responding to nerve injury [31,32,33,34,35,36]. An approach based on the RiboTag method [37,38] has made it possible to compare axonal ribosome-bound mRNAs, from axons of retinal ganglion cells in both developing and adult mice [31]. The identification of a complex translatome in adult axons, encoding proteins with a role in axon survival, neurotransmission, and neurodegenerative disease, provides compelling evidence for an important role of axonal translation in synapse function and maintenance in vivo [39]. The translocation of RBPMS from the soma to the axon in RGCs after an insult suggest that this protein could play a role in post-translational modifications in axons, as RBPs are often necessary to dissociate the ribonucleoprotein (RNP) complex and relieve translational repression of localized transcripts in axons.

In conclusion (Figure 7), the results of this study showed that RBPMS have different distribution between control retinas, hypoxic, and degenerative retinas (explants). These results suggest that after an insult RBPMS changes its distribution as a response to the damage. Further studies are needed in order to elucidate the role of RBPMS under different stress conditions. These characteristic distributions of RBPMS in RGCs suggest that RBPMS could be an excellent marker for RGCs during degeneration.

## 4. Materials and Methods

### 4.1. Eye Samples

Adult porcine eyes (1-year-old) (n = 10) and neonatal eyes (P3) (n = 4) were obtained from a local slaughterhouse and transported to the laboratory in cold CO_2_-independent medium (Life Technologies, Carlsbad, CA, USA) plus 0.1% gentamicin.

Adult rat eyes (n = 10) were obtained from animals that were housed under a 12 h light–dark cycle with ad libitum access to food and water and were humanely sacrificed by exposure to CO_2_.

Donor human eyes (n = 3) were obtained within 24 h postmortem and all donated material was free of any known retinal pathology and contained no obvious ocular trauma or undiagnosed retinal injury at the time of tissue isolation.

Animal experimentation adhered to the ARVO Statement for the Use of Animals in Ophthalmic and Vision Research. Moreover, all the experimental protocols complied with the European (2010/63/UE) and Spanish (RD53/2013) regulations regarding the protection of experimental animals, and they were approved by the Ethics Committee for Animal Welfare at the University of the Basque Country.

### 4.2. Tissue Collection and Cultures

To obtain sections, rat (n = 10), pig eyes (n = 5), and human eyes (n = 3) that were obtained within 24 h postmortem were extracted. One pig eye and one rat eye, and the human eyes were immediately fixed overnight in 4% paraformaldehyde (PFA) and the contralateral eyes were kept for 24 h in a CO_2-_independent medium at 4 °C as a model of hypoxia, and then fixed overnight in 4% PFA. Eyes were cryoprotected for 24 h in 30% sucrose in 0.1 M phosphate buffer at 4 °C and embedded in an optimal cutting temperature (OCT) medium. Cryosections (14 μm thick) were obtained and stored at −20 °C.

To obtain the whole-mount retina, the eyes were dissected, and after removing the lens and the vitreous, the entire retina was isolated. The retinas were fixed in 4% PFA.

To obtain organotypic retinal cultures (explants), under aseptic conditions, each porcine eyeball (n = 3) was immersed in 70% ethanol and then washed in a clean CO_2_-independent medium. The eyes were dissected to remove the lens and the vitreous humor. The entire retina was isolated. Five retinal explants from the middle part of the retina, at the same distance from the optic nerve head and excluding the larger vessels, were obtained from each eye using an 8 mm diameter dissecting trephine. The explants were transferred to cell culture inserts (0.45 μm pore, 12 mm diameter; Merck Millipore, Darmstadt, Germany) with the photoreceptor layer facing the membrane. Explants were cultured in Neurobasal A with 1% L-glutamine and 0.1% gentamicin. Explants were maintained at 37 °C in a humid atmosphere of 5% CO_2_ for 24 h. The culture medium level was maintained in contact with the support membrane beneath the explant and changed with freshly prepared medium on every second day. After overnight fixation, in 4% paraformaldehyde, the explants were cryoprotected overnight in 30% sucrose in 0.1 M PB (phosphate buffer, pH 7.4) at 4 °C, and embedded in an OCT medium. Thereafter, 14 µm cryosections were cut and stored at −20 °C.

### 4.3. Immunochemistry

RGCs were immunostained in whole-mounted retinas and in cryostat sections of the eye. The immunohistochemical techniques were performed at the same time for all species studied. The whole-mounted retinas were blocked with phosphate buffered saline, pH 7.4 (PBS) containing TX-100 (0.25%) overnight, prior to incubation in the primary antibody. The sections were washed twice with PBS-TX-100 for 10 min, and they were then incubated overnight with a primary guinea pig antibody or rabbit anti-RBPMS antibody (1:1000; Phosphosolutions, Aurora, CO, USA), and goat anti-Brn3a (1:1.000; Santa Cruz Biotechnology, Dallas, TX, USA) to detect RGCs, and primary mouse anti-Neurofilament 200 antibody (1:200; Vector Laboratories, Burlingame, CA, USA) to label neuronal axons. After two washes with PBS, antibody binding was detected for 1 h (5 h for whole-mount retinas) with Alexa Fluor 555 goat anti-guinea pig and Alexa Fluor 488 goat anti-mouse secondary antibodies, or Alexa Fluor 555 donkey anti-rabbit and Alexa Fluor 488 goat anti-goat secondary antibodies (Invitrogen, Carlsbad, CA, USA) diluted 1:1000 in PBS-BSA (1%). The sections were washed twice with PBS for 10 min and mounted with a coverslip in PBS:Glycerol (1:1).

### 4.4. Image Capture

Images were acquired with a digital camera (Zeiss Axiocam MRM, Zeiss, Jena, Germany) coupled to an epifluorescence microscope (Zeiss) using the Zen software (Zeiss). The images were photographed at same exposure time, aperture setting, and illumination intensity.

### 4.5. Protein Extraction and Western Blotting

For protein extraction, the samples control (n = 2) and hypoxic (n = 2) retinas were homogenized with RIPA Lysis Buffer (Sigma-Aldrich, St. Louis, MO, USA) and a cocktail of protein inhibitors (Sigma-Aldrich), and sonicated with a Sonopuls Ultrasonic homogenizers (Bandelin). The protein concentration was determined by the Bicinchoninic acid assay. Samples were mixed with the Morris formulation of the SDS-PAGE Sample Buffer (2.5% SDS, 10% glycerol, 5% β-mercaptoethanol) and denatured for 10 min at 98 °C and separated on 10% SDS-polyacrylamide gels (20 and 40 µg/lane). For immunoblotting, proteins were transferred to a nitrocellulose membrane, blocked with 5% bovine serum albumin (BSA) and incubated with primary antibodies: rabbit anti-RBPMS antibody, and in parallel rabbit anti-β-actin antibody was used as internal control overnight at 4 °C. After incubation with the secondary antibody, the blotting membrane was developed with Luminata™ Crescendo Western HRP Substrate (Millipore, Burlington, MA, USA) in a G: BOX (Syngene, Cambridge, UK) and the images were obtained with the GeneSnap Software and analyzed with ImageJ software (version 1.4.3.67) (NIH, Bethesda, MD, USA).

## Figures and Tables

**Figure 1 ijms-21-09330-f001:**
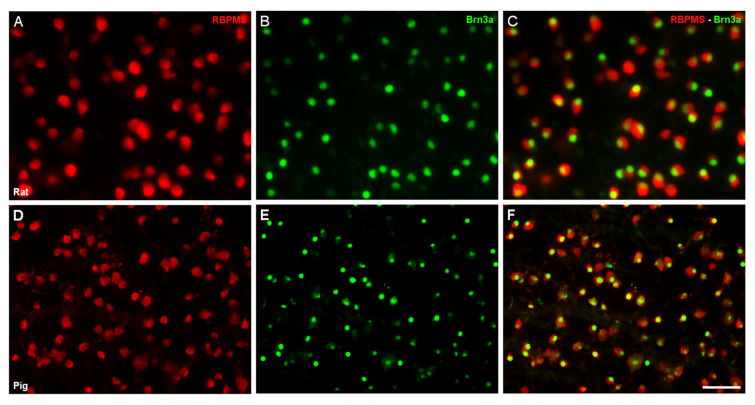
Expression of RNA binding protein with multiple splicing (RBPMS) and Brn3a in pig and rat whole-mount retinas. Retinal ganglion cells (RGCs) were labeled with antibodies against RBPMS (red) (**A**,**D**) and the nuclei of RGCs were labeled with antibodies against Brn3a (green) (**B**,**E**), in rat (**A**–**C**) and pig (**D**–**F**) retinas. Notice that some RGCs are only labeled with RBPMS and not with Brn3a in the merged image (**C**,**F**). Scale bar: 50 µm.

**Figure 2 ijms-21-09330-f002:**
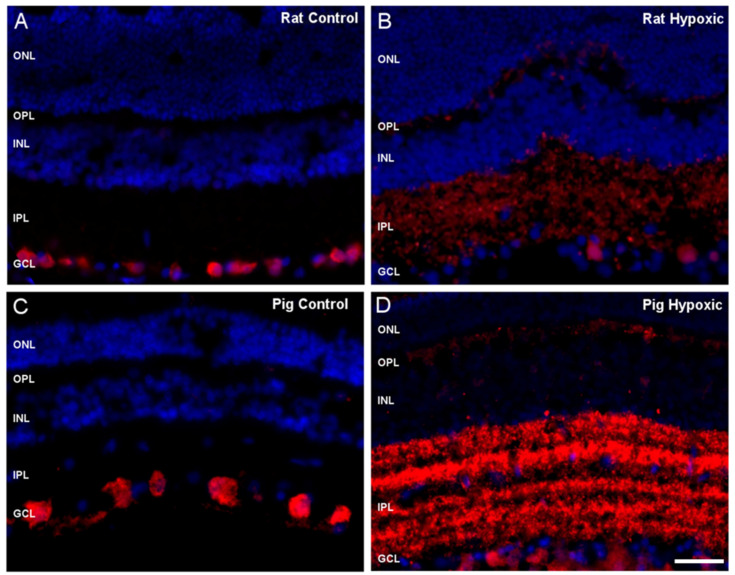
Expression of RBPMS in pig and rat retinal sections under hypoxia. RGCs were labeled with antibodies against RBPMS (red) and the nuclei of retinal cells were labeled with 4′,6-Diamidino-2-Phenylindole (DAPI (blue)). Retinal sections of (**A**) control rat retina, (**B**) hypoxic rat retina, (**C**) control adult pig retina, and (**D**) hypoxic pig retinas were analyzed. Notice the different localization of RBPMS among the samples: in the somas of RGCs in control retinas from (**A**) rats and (**C**) pigs, and in the inner plexiform layer (IPL) and weak labeled in the somas of hypoxic (**B**) rat and (**D**) pig retinas. ONL: outer nuclear layer, OPL: outer plexiform layer, INL: inner nuclear layer, IPL: inner plexiform layer, GCL: ganglion cell layer, NFL: nerve fiber layer. Scale bar: 50 µm.

**Figure 3 ijms-21-09330-f003:**
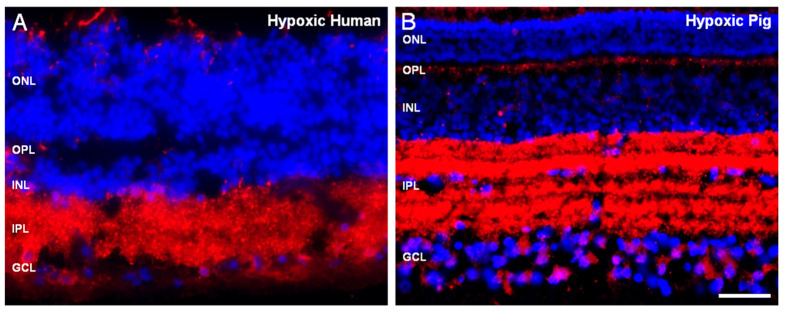
Expression of RBPMS of adult pig and human retinal sections. Retinal sections of (**A**) human retina and (**B**) pig hypoxic retina. RGCs were labeled with the antibody against RBPMS (red) and the nuclei were labeled with DAPI (blue). Notice that in both sections the localization of RBPMS is in the inner plexiform layer (IPL) corresponding to the dendrites and gently labeled in the somas of RGCs. ONL: outer nuclear layer, OPL: outer plexiform layer, INL: inner nuclear layer, IPL: inner plexiform layer, GCL: ganglion cell layer, NFL: nerve fiber layer. Scale bar: 50 µm.

**Figure 4 ijms-21-09330-f004:**
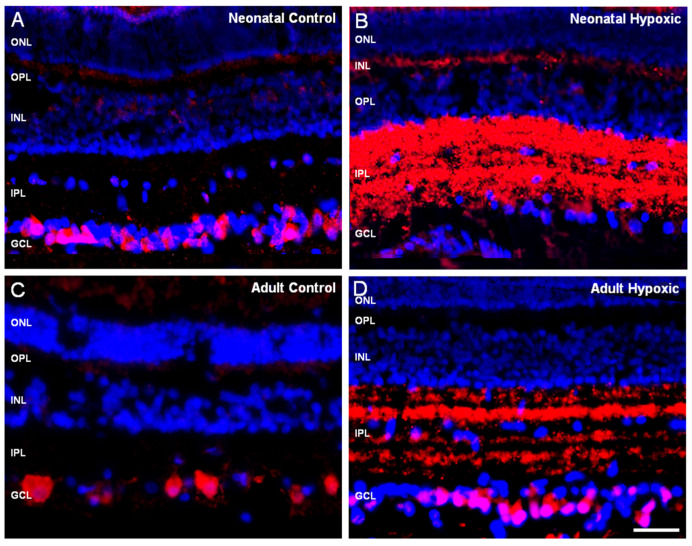
Expression of RBPMS of adult and neonatal pig retinal sections in control and hypoxic conditions. Retinal sections of (**A**) control neonatal pig retina, (**B**) neonatal pig hypoxic retina, (**C**) adult control retina, and (**D**) adult hypoxic retina. RGCs were labeled with the antibody against RBPMS (red) and the nuclei were labeled with DAPI (blue). Notice that in control retinas, the expression of RBPMS is in the somas of RGCs (**A**,**C**) and in both hypoxic sections the localization of RBPMS was in the IPL corresponding to the dendrites and gently labeled in the somas of RGCs (**B**,**D**). ONL: outer nuclear layer, OPL: outer plexiform layer, INL: inner nuclear layer, IPL: inner plexiform layer, GCL: ganglion cell layer, NFL: nerve fiber layer. Scale bar: 50 µm.

**Figure 5 ijms-21-09330-f005:**
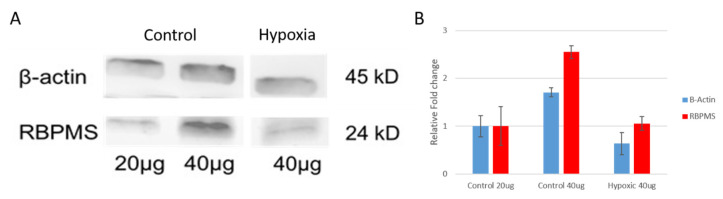
Representative Western Blot analysis of pig control and hypoxic protein retinal extractions. Western Blot analysis that shows the expression level of RBPMS in pig control retinas at different concentrations of protein (20 and 40 µg) and from hypoxic pig retinas (40 µg of protein). (**A**) β-actin was used as endogenous control. The graph represents the relative fold change from mean values obtained from the (**B**) optic densities of two independent experiments. Notice that the band of 20 µg of control pig retinas were similar to the 40 µg band of the hypoxic retinas.

**Figure 6 ijms-21-09330-f006:**
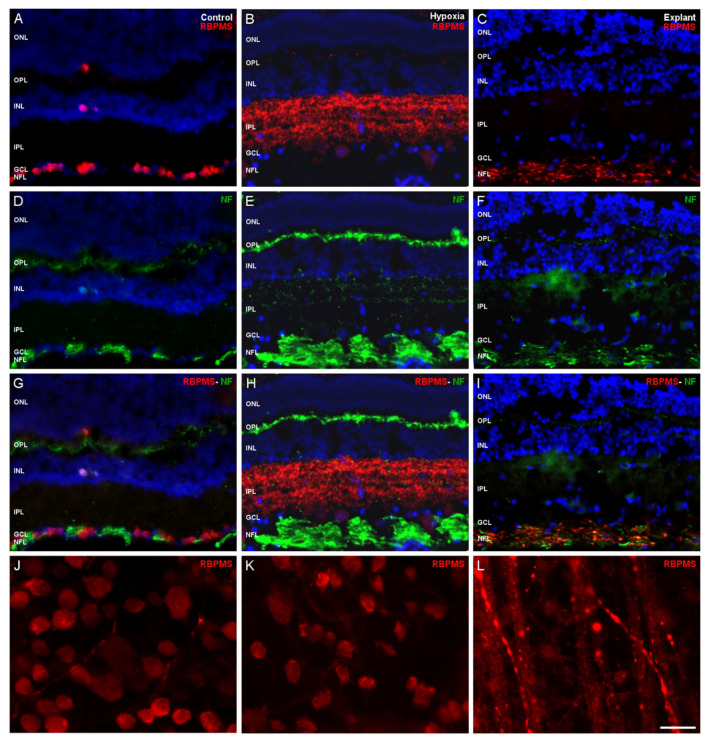
Expression of RBPMS and neurofilament (NF) in adult pig retinal organotypic cultures compared to control and hypoxic retinas. (**A**,**D**,**G**) Retinal section from control, (**B**,**E**,**H**) hypoxic, (**C**,**F**,**I**) explants, and whole-mount retina from (**J**) control, (**K**) hypoxic, and (**L**) explants from adult pig retinas at 1 day of culture were represented. (**A**–**C**,**J**–**L**) RGCs were labeled with the antibody against RBPMS (red), (**D**–**I**) the neuronal axons were labeled with the antibody against NF (green), and the nuclei were labeled with DAPI (blue). (**G**–**I**) Merge images were also represented. Notice that RBPMS labeling changes their localization from the somas of RGCs in (**A**) control, to the inner nuclear layer (INL) in (**B**) hypoxia, to nerve fiber layer (NFL) during the degeneration of the (**C**) explants corresponding to the axons of RGCs as indicated by the (**D**–**F**) co-labeling with NF. This acquired a punctuated look into the axons’ length as we can see in the (**L**) whole-mount retinal explant. In the whole-mount retinas, RGCs are labeled with the (**J**–**L**) antibody against RBPMS (red). ONL: outer nuclear layer, OPL: outer plexiform layer, INL: inner nuclear layer, IPL: inner plexiform layer, GCL: ganglion cell layer, NFL: nerve fiber layer. Scale bar: 50 µm.

**Figure 7 ijms-21-09330-f007:**
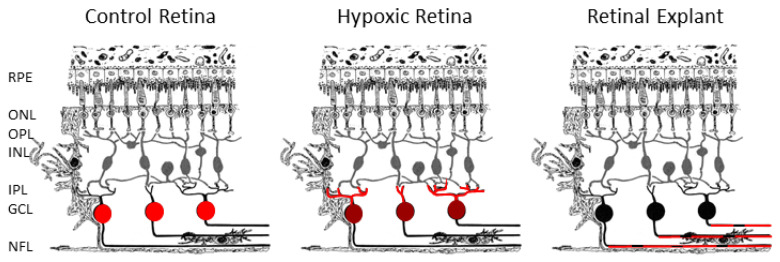
Scheme showing the expression of RBPMS in control, hypoxic, and in retinal explants. RBPMS (red) is expressed in the somas of RGCs in control retinas, in dendrites in the IPL, and more lightly, in the somas in hypoxic retinas and in axons in retinal explants. ONL: outer nuclear layer, OPL: outer plexiform layer, INL: inner nuclear layer, IPL: inner plexiform layer, GCL: ganglion cell layer, NFL: nerve fiber layer.

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
