# Peer review of "Differential Distribution of RBPMS in Pig, Rat, and Human Retina after Damage"

_ijms, 2020, doi:10.3390/ijms21239330_

Round 1
Reviewer 1 Report
The study by Pereiro et al. examines the expression and localization of the RNA binding protein with multiple splicing (RBPMS) in rat, pig, and human retinas, both developing and mature, and in physiological as well as hypoxic conditions. RBPMS is primarily found in the inner retina, specifically in the soma of retinal ganglion cells (RGC), but its function is currently unknown. During hypoxia, this team shows that while RBPMS levels decrease in RGC soma, they increase substantially in RGC dendrites in the inner plexiform layer. Intriguingly, in retinal explants generated after optic nerve axotomy, RBPMS is observed primarily in degenerating RGC axons in the nerve fiber layer. These findings suggest that RBPMS is transported to different RGC compartments after injury and could potentially contribute to RNA processing, stability, and transport.
This study is well-written and the experiments are carefully done and include appropriate controls. The data support the conclusions drawn by the authors. The findings are interesting and provide new insights into a RGC-specific protein, whose role is not well understood, and might contribute to the RGC response to hypoxic damage. Given that RBPMS has become a popular and widely used RGC marker, and possibly the most broadly expressed protein among different RGC subtypes, it is of great value to understand RBPMS changes in naïve and injured RGC. In addition, dendritic and axonal degeneration are major features of RGC loss in optic neuropathies such as glaucoma, thus it is important to understand changes in these RGC compartments. There are a few issues that, if addressed, would further strengthen the manuscript.
- In the results, pages 75-77, the authors indicate: “However, under hypoxic conditions the RBPMS was expressed in the inner plexiform layer (IPL), whereas the RGCs had reduced labeling of RBPMS in the somas (Figure 2 B, D) in both pig and rat retinas.” This is the first time that hypoxia is mentioned in the results and it would be useful to indicate how they induced hypoxia. This is described in the Methods, but it would be useful for the reader to readily have this information in the Results as well.
- Along the same lines, in lines 88-90, they indicate that: “RBPMS was expressed in the INL in human retinal sections and reduced in the somas of RGCs, a pattern similar to pig and rat retinas after hypoxia (Figure 3). This fact could be due to the human eyes were obtained 24 hours post mortem.” It would be helpful to state more clearly that human retinas collected at 24 hours post-mortem are likely to be hypoxic relative to retinas collected immediately after death (controls). Of note, this is an assumption, and should be stated as such, given that real oxygen levels in these retinas were not measured.
- Do the authors expect RGC death at 24 hours of hypoxia? Given the reduction in RBPMS staining in the RGC soma, please explain how you rule out that this is not due to RGC loss.
- In figure 5, western blot analysis shows RBPMS levels in pig control retinas at different protein concentrations (20 µg and 40 µg) and hypoxic pig retinas (40 µg of protein). B-actin was used as endogenous control. The authors indicate that: “It should be noted that the bands of 20µg of protein of control is similar to the band of 40µg protein of hypoxic retinas consequently, it seems that there is increased expression of RBPMS in control retinas.” The rationale for making this statement is not clear. Indeed, the RBPMS-positive bands at 20 µg (control) and 40 µg (hypoxia) are similar, but so are the corresponding bands probed with the B-actin control (or at least they seem similar in the image provided). B-actin is supposed to be a control for protein loading. If the bands are similar, it indicates that they carry the same amount of protein. Please clarify.
- An interesting, but intriguing finding, is the lack of RBPMS expression in the IPL in retinal explants. Instead, the authors observed RBPMS accumulation in RGC axons. However, retinal explants are prepared from axotomized retinas, which are also presumably hypoxic because the central retinal artery is cut along with the optic nerve. As far as I could see from the methods, the retinal explants are not oxygenated, so they could also be hypoxic (potentially). Please explain how you rule out the potential role of hypoxia in this response and why under these conditions there is no RBPMS accumulation dendrites. Is it possible that different injury modalities (hypoxia versus axotomy) induce distinct distribution of RBPMS to separate RGC cellular compartments? Please discuss.
Minor comments:
- In the abstract, last sentence: “Therefore, RBPMS should be further investigated in relation to its role in axonal degeneration.” Please change to: “Therefore, RBPMS should be further investigated in relation to its role in axonal and dendritic degeneration.
- Please replace “zebra fish” for “zebrafish” throughout the text.
- Line 45, change “…maturing neurons and it function in RGC axons” to “…maturing neurons and its function in RGC axons.
- Line 48, change “its function in the mammal retinas” for “its function in the mammalian retinas”.
- Scale bars are missing in Figure 2.
Reviewer 2 Report
The manuscript “Differential distribution of RBPMS in pig, rat and human retina after damage” by Xandra Pereiro and co-authors evaluates RBMPS as a retinal ganglion cell marker in different species. They compare pig, rat, and human retina. The manuscript is well written, but some improvements could be made.
Introduction
- In line 49: why “on the other hand”? I don’t really see the connection to the previous paragraph
- The authors should give a little bit more information about the three different species used (similarities and differences in regard to the retina)
Methods
- Please also add the number of eyes for rats and humans to 4.1. Eye samples
- 3. Immunochemistry: was the same protocol used for all species?
- Were eyes from the different species fixed differently?
- Please clarify which methods were used from which species and how many eyes/retinas were used for each method.
Results
- The authors need to provide RBPMS and Brn-3a cell counts to confirm the stated similarities for both markers (fig. 1).
- Fig 1c: to looks like some RBPMS+ cells were not Brn-3a+ as stated by the authors. How could this be explained?
- Figure 2: scale bar is missing.
- Line 112: this sentence is hard to understand, please rewrite it.
- The authors show exemplary Western blot bands ,where these quantified?
- Figure 6: is there a DAPI signal in J-L? in general the signal is hard to see in these flatmount pictures.
Round 2
Reviewer 2 Report
The authors revised the manuscript and addressed all previous concerns.